# Computer-Assisted Differentiation between Colon-Mesocolon and Retroperitoneum Using Hyperspectral Imaging (HSI) Technology

**DOI:** 10.3390/diagnostics12092225

**Published:** 2022-09-15

**Authors:** Nariaki Okamoto, María Rita Rodríguez-Luna, Valentin Bencteux, Mahdi Al-Taher, Lorenzo Cinelli, Eric Felli, Takeshi Urade, Richard Nkusi, Didier Mutter, Jacques Marescaux, Alexandre Hostettler, Toby Collins, Michele Diana

**Affiliations:** 1Research Institute against Digestive Cancer (IRCAD), 67091 Strasbourg, France; 2ICube Laboratory, Photonics Instrumentation for Health, 67081 Strasbourg, France; 3Department of Surgery, Maastricht University Medical Center, 6229 ER Maastricht, The Netherlands; 4Department of Gastrointestinal Surgery, San Raffaele Hospital IRCCS, 20132 Milan, Italy; 5Department of Surgery, Division of Hepato-Biliary-Pancreatic Surgery, Kobe University Graduate School of Medicine, Kobe 6500017, Japan; 6Research Institute against Digestive Cancer (IRCAD), Kigali, Rwanda; 7Department of Digestive and Endocrine Surgery, Nouvel Hôpital Civil, University of Strasbourg, 67091 Strasbourg, France; 8IHU-Strasbourg—Institut de Chirurgie Guidée par L’image, 67091 Strasbourg, France

**Keywords:** hyperspectral imaging, artificial intelligence, intraoperative navigation tool, optical imaging, deep learning, convolutional neural network, colorectal surgery

## Abstract

Complete mesocolic excision (CME), which involves the adequate resection of the tumor-bearing colonic segment with “en bloc” removal of its mesocolon along embryological fascial planes is associated with superior oncological outcomes. However, CME presents a higher complication rate compared to non-CME resections due to a higher risk of vascular injury. Hyperspectral imaging (HSI) is a contrast-free optical imaging technology, which facilitates the quantitative imaging of physiological tissue parameters and the visualization of anatomical structures. This study evaluates the accuracy of HSI combined with deep learning (DL) to differentiate the colon and its mesenteric tissue from retroperitoneal tissue. In an animal study including 20 pig models, intraoperative hyperspectral images of the sigmoid colon, sigmoid mesentery, and retroperitoneum were recorded. A convolutional neural network (CNN) was trained to distinguish the two tissue classes using HSI data, validated with a leave-one-out cross-validation process. The overall recognition sensitivity of the tissues to be preserved (retroperitoneum) and the tissues to be resected (colon and mesentery) was 79.0 ± 21.0% and 86.0 ± 16.0%, respectively. Automatic classification based on HSI and CNNs is a promising tool to automatically, non-invasively, and objectively differentiate the colon and its mesentery from retroperitoneal tissue.

## 1. Introduction

Colorectal cancer (CRC) is the third most prevalent cancer with the second highest mortality rate worldwide [1]. Over the past few decades, the management of CRC has improved significantly due to better screening policies and the introduction of extended radical “en bloc” surgical resection such as complete mesocolic excision (CME) [2]. Hohenberger proposed CME in colonic resection in 2009 [3]. CME consists of the substantial resection of tumor-bearing bowel segments with “en bloc” removal of the mesocolon along embryological fascial planes. Although CME has been associated with adverse outcomes, such as a greater intraoperative blood loss and a higher incidence of postoperative surgical complications [4], several studies have shown that CME has superior oncological outcomes [3,5,6,7,8]. The number of lymph nodes harvested in oncological resection represents an independent prognostic factor for survival [9,10,11], and CME specimens are generally higher quality with a higher number of lymph nodes compared to non-CME resections, which may account for the lower risk of local recurrence [12].

A sound understanding of colorectal anatomy is crucial to the correct performance of CME and prevention of surgical complications. Culligan et al. [13] precisely described the fusion fascia, which is located between the mesentery, the retroperitoneum, and a surgical exfoliation layer that is formed within the fusion fascia. The authors demonstrated that the surgical exfoliation layer can be deliberately dissected when mobilizing the colon and its mesentery. This extensive dissection between the mesentery and the retroperitoneum in CME is generally performed by visually assessing differences in the microvasculature, which is subjective and requires a high level of expertise. In addition, in some cases, it is challenging to recognize these subtle patterns with the naked eye or with white-light laparoscopy. These technical details represent major obstacles to the identification of adequate cleavage and dissection planes during CME.

In a systematic review including 18,989 patients comparing CME with conventional colorectal resection for colorectal cancer, CME was associated with improved 3- and 5-year overall survival, improved 3-year disease-free survival, and decreased local and distant recurrences [14,15]. The authors reported that CME does not increase the risk of postoperative mortality or anastomotic leakage. However, CME was associated with an increased risk of postoperative complications such as splenic and superior mesenteric vein injuries. Additionally, structures behind the peritoneum that are referred to as “retroperitoneal”, such as the ureter and gonadal vessels, could be injured [16]. Ureteral injury is a serious complication during intrapelvic surgery, and the majority of iatrogenic ureteral injuries can lead to severe morbidity and even mortality [17,18]. The intraoperative identification of anatomical layers can protect organs from iatrogenic damage during colorectal surgery. Ongoing research has been devoted to refining intraoperative methods to confirm the anatomical position of these structures, especially the ureter. Technologies include near-infrared fluorescence imaging (NIRF) using indocyanine green (ICG) or methylene blue (MB). However, they provide suboptimal ureteral visualization in comparison to novel fluorescent dyes such as CW800-BK and ZW800-1, which are currently undergoing clinical translation [19,20,21,22,23,24,25,26]. Therefore, ureteral injury still represents a burden.

Over the past few decades, surgeons have made use of new optical imaging technologies to enhance their vision in digestive surgery, especially during colorectal resections. This is reflected by a large number of studies on intraoperative NIRF surgery, which have been used to characterize tumors [27], evaluate perfusion at the level of anastomoses [28,29,30,31], identify sentinel lymph nodes [32], and visualize real-time lymphatic flow intraoperatively [33,34]. The advantages of NIRF-guided surgery include its high sensitivity, rapid feedback, and lack of radiation [34]. However, the narrow range of optimal concentrations makes it difficult to improve administration protocols (concentration, dose, and timing). In addition, the low penetration of the NIR-I window (wavelength of 700–900 nm in the electromagnetic spectrum) [35] makes it challenging to visualize deeper anatomical structures such as the ureter. Fluorescence guidance using ICG lymphadenectomy in CME requires endoscopic submucosal injection into the proximity of the tumor for at least 12 h prior to the operation for precise intraoperative mapping [36]. Subserosal dye injection has also been studied to intraoperatively demonstrate lymphatic drainage [37]. However, one of the greatest drawbacks of ICG is its water-soluble nature; as it diffuses over time, the ability to accurately locate affected areas is reduced.

Hyperspectral imaging (HSI) is a contrast-free optical imaging technology, which combines a photographic camera and a spectroscope [38]. Contrarily to exogenous fluorescence (e.g., ICG and MB), HSI provides intraoperative and contactless quantitative imaging of intrinsic physiological properties [39], including tissue oxygenation, tumor identification [40,41], organ perfusion assessment [42,43], and identification of key anatomical structures. Machine learning and deep learning, combined with optical imaging, are gaining popularity in the medical field to support surgical decision making using large training datasets. Artificial intelligence (AI) and computer vision have improved computer-assisted diagnosis by recognizing dysplastic/neoplastic polyps in image-guided surgery through the detection of critical steps during minimally invasive procedures such as endoscopic sleeve gastrectomy [44]. At present, AI algorithms are frequently developed using standard color images with three optical channels of red, blue, and green (RGB), which have significant limitations, such as a lack of quantitative parameters. AI-based automatic recognition of colorectal tumors is still in its infancy, and quantitative optical imaging modalities such as HSI could help to further advance AI algorithms towards a higher accuracy and precision. The combination of HSI and AI has recently generated promising results. Using a four-layer neural network, Boris Jansen-Winkeln et al. achieved an 86% sensitivity and a 95% specificity in identifying colorectal cancer (CRC) [45]. Barberio et al. showed that HSI combined with CNNs could be used to automatically recognize key anatomical structures such as blood vessels and nerves [46]. Collins and Maktabi [47] showed that colorectal and esophagogastric cancer could also be detected with various machine learning models, and CNNs often produced the best results.

The aim of this study was to assess the accuracy of HSI technology in combination with CNNs to automatically distinguish colonic and mesenteric tissue (which are to be resected in CME) from retroperitoneal tissue (which is to be spared).

## 2. Materials and Methods

### 2.1. Animals

All experiments were performed at the Research Institute against Digestive Cancer (IRCAD, France). A total of 20 adult pig models (*Sus scrofa domesticus*, ssp. large white, mean weight: 40 kg) were included. The study was part of the ELIOS protocol (Endoscopic Luminescent Imaging for Oncology Surgery), fully approved by the local Ethical Committee on Animal Experimentation (ICOMETH No. 38.2016.01.085) and by the French Ministry of Superior Education and Research (MESR) (APAFIS#8721-2017013010316298-v2). All animals used in the experimental laboratory were managed according to French laws for animal use and care and according to the directives of the European Community Council (2010/63/EU) and ARRIVE guidelines [48]. The animals were housed and acclimatized for 48 h in an enriched environment, respecting circadian cycles of light/darkness, and with constant humidity and temperature conditions. They were fasted 24 h before surgery, with ad libitum access to water, and finally sedated (zolazepam + tiletamine 10 mg/kg IM) 30 min before the procedure in order to decrease stress. Anesthesia induction was administered with propofol (3 mg/kg) injected intravenously (18 G IV catheter in an ear vein). The animals were maintained with rocuronium (0.8 mg/kg) along with inhaled isoflurane 2%. At the end of the protocol, animals were euthanized with a lethal dose of pentobarbital (40 mg/kg).

### 2.2. Anatomical Relevance, Surgical Procedure, and Hyperspectral Data Acquisition

Anatomical structures and their embryological phylogeny should be understood during surgery. The mesentery and the retroperitoneum are derived from the same embryological structure referred to as mesoderm, and they are connected to the abdominal cavity. The mesentery is a double layer of peritoneum, which includes retroperitoneum, connective tissue, blood vessels, nerves, lymph nodes, and adipose tissue [49]. The spectral profiles of the mesentery and retroperitoneum differ based on several parameters, including water content (peaking at about 980 nm) and adipose content (peaking at about 740 nm) [50]. In addition, the spectral profile of hemoglobin (Hb) significantly differs between the oxygenated and deoxygenated states and strongly contributes to the overall tissue spectral profile [51]. Changes in hemoglobin concentration and oxygen saturation due to vascular content can be measured using HSI.

To expose intraperitoneal and retroperitoneal structures, the abdominal cavity was accessed via a midline laparotomy, and a self-retaining retractor was placed to expose the region of interest (ROI).

HSI images were obtained from the right or left side of the sigmoid colon mesentery at the level of the inferior mesenteric artery, including the sigmoid colon and the retroperitoneum. The HSI camera (TIVITA^®^, Diaspective Vision GmbH, Germany) was a push-broom scanning device with a complementary metal oxide semiconductor (CMOS) image sensor with a spatial resolution of 640 × 476 pixels and a spectral range from 500 to 1000 nm (5 nm spectral resolution increments, totaling 100 bins). In the first increment, 501 wavelength bands were used from 500 to 1000 nm. In a second increment, 5× binning in the wavelength dimension was performed, resulting in a final hypercube of 100 frequency bins, namely the wavelength from 500 to 995 nm. However, due to binning settings, the wavelengths from 996 to 999 were included.

The HSI camera was positioned 50 cm above the ROI and, immediately before and during image capture, environmental lights were turned off, and ventilation was paused. Capture time took approximately 6 s, and the placement of the camera during the procedure is represented in Figure 1. In this study, relative values were considered, and no white balance was performed during image acquisitions.

The optical system causes a significant smile on the camera sensor. A significant keystone effect is not observable if the spectrometer is properly adjusted. Wavelength calibration is performed for every single row of the raw sensor image. With this method, the wavelength calibration and the smile correction are performed in one single step. The result is a wavelength image with the y and the calibrated λ-dimension from 500 to 1000 nm. Every single spectrometer unit is tested for correct imaging of the spectral lines of the krypton gas lamp during and after production. This principle was proven with different spectral measurement standards and substances where spectra are well-known (e.g., colorchecker, use of different liquids, etc.). The calibrated HSI data were used to develop and train the CNN.

### 2.3. Deep Learning Model

Based on a previous animal study from Barberio et al. comparing the two most successful deep learning models, i.e., support vector machines (SVMs) [52,53,54] and CNNs for HSI tissue segmentation, the CNN model achieved an overall higher sensitivity for all tissue classes (89.4%) except for the nerve class, which had a sensitivity of 76.3% [46]. Consequently, this combined with deep learning allowed for the automatic discrimination of six different tissue classes (artery, vein, adipose, muscle, skin, and nerve), supporting our decision to adopt the CNN model in this study.

### 2.4. Annotation

Immediately after each image acquisition, the operating surgeons (N.O and M.R.R.L) used an image manipulation software (GIMP, GNU Image Manipulation Program, open source) to manually annotate the RGB images associated with each HSI (Figure 2A). Figure 2B shows examples of annotated images visualized in grayscale with overlaid annotations represented as colored regions. Annotations in green represent tissue to be removed (colon and mesocolon), and purple annotations were used for the tissue to be preserved (retroperitoneum). The total number of annotated colon-mesocolon pixels was 357,811, with an average of 17,890 ± 12,438 (SD) per image. The total number of annotated peritoneum pixels was 81,655, with an average of 4083 ± 3749 (SD) per image. Consequently, the data had a class imbalance of 4.32:1.

A CNN was then trained to distinguish colon and mesocolon tissue from retroperitoneal tissue using deep learning.

### 2.5. CNN Model Training and Evaluation

#### 2.5.1. Image Processing Pipeline with a Trained CNN

For each pixel of interest in an HSI, one HSI sub-volume was extracted and centered on the corresponding spatial coordinates of the pixel. This sub-volume was then used to train a CNN, which generated a predictive score for the two tissue classes, and the tissue class with the highest score was associated with the pixel. This process was then repeated for each pixel, generating a spatial tissue prediction map (also known as image segmentation).

#### 2.5.2. CNN Architecture

CNNs have been the dominant machine learning model for pattern recognition in optical image data, including HSI [55,56]. A CNN learns relevant spatiospectral features through a series of trainable convolutional filters (hidden layers). The extracted features are fed to the following layer in order to generate a hierarchy of spatiospectral features. The filter weights are trainable parameters, which are automatically adjusted during training. Deeper hidden layers are used to extract higher level features. Based on the extracted features, the last layer determines the classification prediction.

The CNN architecture that we selected [57] has been shown to work well on related tissue recognition problems [46,47], and its architecture is provided in Table 1. The CNN used a spatial window of 5 by 5 pixels and had 31,532 trainable parameters with seven hidden layers, including six convolutional layers with ReLU activations. It had one fully connected final layer with two output neurons corresponding to two classes; the first class was colon or mesocolon tissue (colon-mesocolon), and the second class was retroperitoneal tissue. Class imbalance was handled in the training loss function by downweighing the majority class using inverse frequency median weighting [58].

#### 2.5.3. CNN Training

HSI sub-volumes were centered on every annotated hyperspectral image pixel, forming spatial patches of 5 by 5 pixels, corresponding to 5 by 5 by 100 sub-volumes. The third sub-volume dimension corresponded to the spectral dimension, which represented 100 spectral wavelengths produced by the camera. In total, there were 439,466 training samples. Because this study considered 20 images, a fixed train/validate/test split (typically using 70%/30%/10%) was not possible (only two images for testing). As a result, the model’s ability to generalize on independent (untouched) data was checked with k-fold cross-validation, and special care was taken to prevent data leakage (occurring when a model has access to information in the test set, which could artificially inflate its performance). To maintain test independence, no information about the test data was ever used to influence the trained CNN: the CNN’s design and settings (e.g., spatial window size, structure, activation functions, etc.) and the training parameters (e.g., learning rate, weight decay, and optimization algorithm) were all programmed before data were collected. Consequently, there was a strong isolation between model development and performance evaluation. The CNN design was the same as used in our prior animal study on HSI-based tissue recognition [46] and our human study on HSI-based colorectal cancer recognition [59]. All training parameters were similar to Collins et al.’s [59], and they were not modified in this study’s data.

To train and test the CNN, the annotated sub-volume samples were split into training and test sets using 5-fold cross-validation (CV). Importantly, this was performed so that the CNN was never trained and tested on data from the same animal. The 20 hyperspectral images (one per animal) were randomly partitioned into five sets (S1, …, S5), with each set having four HSIs. Five CNN models were then trained. The first CNN was trained using the annotated HSI sub-volumes in sets S1, S2, S3, and S4, (16 images); its performance was then tested on the annotated sub-volumes in set S5 (four images). Four other CNN models were trained similarly, each using a different test set that was excluded from its training set. The CNNs were trained using batch gradient descent with binary cross-entropy loss function and inverse median frequency class balancing. Training was run for 250 epochs using a batch size of 8192, a learning rate of 0.01, and a weight decay of 0.0005. The CNNs were implemented using PyTorch v1.4.

### 2.6. Performance Metrics and Statistical Methods

Performance was evaluated with standard metrics used in machine learning, implemented by means of Python scikit-learn (version 0.22.1, https://scikit-learn.org; (accessed on 17 August 2021)). Sensitivity, specificity, and two other well-established performance metrics were used, i.e., the F1 score and the receiver operator curve area under curve (ROC AUC). Unlike sensitivity and specificity, F1 gives a single performance score (the harmonic mean of recall and precision). The ROC AUC is a complementary performance statistic, which shows the model’s ability to rank samples. (It gives a higher score for a true class compared to a false class.) Unlike sensitivity, specificity, and F1, ROC AUC is popular since it indicates the model’s predictive performance without requiring a decision threshold.

## 3. Results

### 3.1. Performance Metrics

The quantitative results are summarized in Table 2. The CNNs achieved a relatively high mean sensitivity for colon-mesocolon (86.0 ± 16.0%) and retroperitoneum (79.0 ± 21.0%). The mean F1 score was 0.90 ± 0.11 for colon-mesocolon and 0.65 ± 0.25 for retroperitoneum. The mean ROC AUC was 0.92 ± 0.12 for colon-mesocolon and 0.92 ± 0.12 for retroperitoneum, both of which are considered outstanding [60].

### 3.2. Performance Visualization

Visually, the CNNs performed differentiation adequately between colon-mesocolon and retroperitoneum with minimal errors. Representative results are shown in Figure 2 with three cases arranged in three columns. In the first column, most of the colon-mesocolon tissue was correctly recognized (green regions) in congruence with the relatively high mean colon-mesocolon sensitivity score of 0.86 (Table 1). Most of the retroperitoneum tissue was correctly recognized (blue regions). However, there was some misclassification with the colon-mesocolon. It was in line with the lower colon-mesocolon specificity of 0.79 (Table 1). When considering only the first image (top row in Figure 2), the colon-mesocolon sensitivity was 0.99, and specificity was 0.80, indicating some colon-mesocolon over-segmentation. The second image (middle row in Figure 2) had a sensitivity of 0.82 and specificity of 0.95, indicating some colon-mesocolon under-segmentation. The third image (bottom row in Figure 2) had a detection sensitivity of 0.95 and specificity of 0.95.

## 4. Discussion

The objective of this study is to demonstrate the ability of the combination of HSI with CNNs trained with deep learning to discriminate between tissues to be resected (colon and mesocolon) from tissues to be preserved (retroperitoneum) during colorectal resection.

The mainstay of image-guided surgery in recent decades has been focused on NIRF. NIRF-guided surgery has had excellent results in improving the visualization of tissue with blood and lymphatic flow [27,28,29,30,31,32,33,34]. However, the recognition of dissection planes necessary for accurate CME surgery is difficult to achieve using NIRF. The challenge in CME is to detect the so-called “avascular plane” in which only microscopic blood vessels pass: this is the focus of ICG IV infusion vascular mapping [61].

Recently, there has been great interest in investigating contrast-free optical imaging modalities. HSI has become a powerful tool to objectively assess the unseen based on the interaction of emitted, absorbed, reflected, and scattered light with biochemical tissue components. This technique, which is contrast-free and non-invasive, has the potential to become a valuable navigation tool and the future of intraoperative guidance [62,63]. However, HSI data are very difficult for the surgeon to directly interpret. Surgeons must have access to advanced machine learning algorithms to process data in order to automatically recognize tissue patterns based on their spatiospectral signatures; this provides surgeons with precise and relevant tissue information, which is not visible to the naked eye or using a standard RGB camera.

In the current in vivo non-survival porcine model, using HSI in combination with a CNN, we were able to automatically recognize the anatomy of the mesothelial layer that needs to be dissected during CME, which constitutes a critical key step during advanced colonic resection.

The potential clinical relevance of HSI lies in the better replicability of CME—assuring removal of all lymph nodes while decreasing the recurrence rate. An HSI-guided dissection could help to achieve more accurate and extensive dissection adhering to embryological planes while preventing damage to other retroperitoneal organs and also preventing neoplastic tissue from being unresected.

Very promising results were obtained in this study. However, there are some limitations. The HSI camera system had a limited electromagnetic spectral range (500–1000 nm), a relatively low spectral resolution (5 nm), and low spatial resolution (640 by 476 pixels). Consequently, the use of low-resolution images could very well decrease accuracy in the annotation process even when performed by experienced hands, which may introduce errors in the dataset. To minimize such errors, two experienced surgeons made annotations in the operating room while verifying the dissection planes intraoperatively. The main reason for using this type of camera was that the spectral profile could be understood at molecular level in this spectral range (500–1000 nm) [35]. It also did not disrupt the surgical workflow (approximately 6 s in acquisition time), and it was clinically approved, making it easier to translate our study results into clinical trials and future research in colorectal surgery.

A second limitation lies in the small number of samples using porcine models, in which the mesentery is significantly thinner as compared to human models, which have more adipose tissue. As a result, it is uncertain whether similar results could be obtained in clinical human studies. In humans, there is a well-described fusion fascia between the retroperitoneum and the mesocolon dissected during CME.

A third limitation concerns class imbalance: there was more colon-mesocolon tissue present in the dataset as compared to retroperitoneum by a factor of 4.32. This was handled during training by inverse frequency median weighting [58], which is an established technique used to decrease the influence of the majority class. This was effective because the average test sensitivity and specificity were similar (0.86 vs. 0.79) although trained on a greater number of colon-mesocolon data. However, the model still had a tendency (approximately 7%) to over-segment colon-mesocolon tissue. In clinical use, it may be necessary to adjust the detection threshold to achieve the desired sensitivity/specificity trade-off, balancing the risk of lymphatic tissue residue and intraoperative complications aimed at improving the precision of CME surgery.

The constraint related to data imbalance includes the annotation phase, which was represented by narrowly annotated retroperitoneal areas. Expert surgeons avoided annotating areas with ascites (inflammatory water content due to laparotomy) and major vascular vessels (i.e., aorta, vena cava) since annotating these areas could well lead to inaccurate results. The lower F1 scores for retroperitoneum as compared to colon-mesocolon tissue were due to the fact that the model had a lower positive predictive value (i.e., precision) in regard to retroperitoneal tissue. This may be explained by the fact that the annotated retroperitoneal tissue often had thin regions (as shown in Figure 2), and F1 (also known as the Dice metric [64,65]) is more sensitive to segmentation errors in thin structures.

Since pig models do not have a clear fusion fascia, this study used HSI to recognize the mesentery and retroperitoneum in the sigmoid colon only before any dissection took place, and the potential effects of energy sources on spectral images were not evaluated. Previously, our group identified that electrocautery resulted in low-level oxygen perfusion while using HSI data in a liver resection model [62]. The peritoneum is a thin mesothelial layer. Consequently, the use of electrocautery could well cause inaccurate recognition during surgical dissection. The next sensible step in the clinical translation of HSI in combination with a CNN as a navigational tool for colorectal surgery is to study whether the system can achieve high sensitivity and specificity in humans. The objective would be twofold: firstly, to recognize the boundary of the mesothelial layer that needs to be dissected and, secondly, to determine whether surgical techniques essential in performing the dissections required in CME affect the accuracy of the recognition system.

Finally, due to the open camera set-up, a laparotomy was required for the acquisition of images. At present, minimally invasive laparoscopic surgery is the standard of care in oncological colorectal resections. Zuzak et al. developed a NIR laparoscopic HSI system prototype to guide laparoscopic liver resection and differentiate the portal vein from the extrahepatic biliary duct and the hepatic artery [27]. Recently, HSI-equipped laparoscopic cameras are becoming commercially available, and the major limitation with the open system in terms of narrowly annotated retroperitoneal areas could be overcome using HSI laparoscopic equipment since the operative setting and field of view may be improved. Consequently, further studies are necessary to establish the accuracy of HSI systems during minimally invasive surgery.

## 5. Conclusions

In this experimental study, HSI has been combined with CNNs and deep learning to accurately distinguish two tissue groups (i.e., the colon and its mesentery from the retroperitoneum). HSI recognition could well play a future role in intraoperative guidance for CME. Additional preclinical and clinical trials are required to corroborate our results.

## Figures and Tables

**Figure 1 diagnostics-12-02225-f001:**
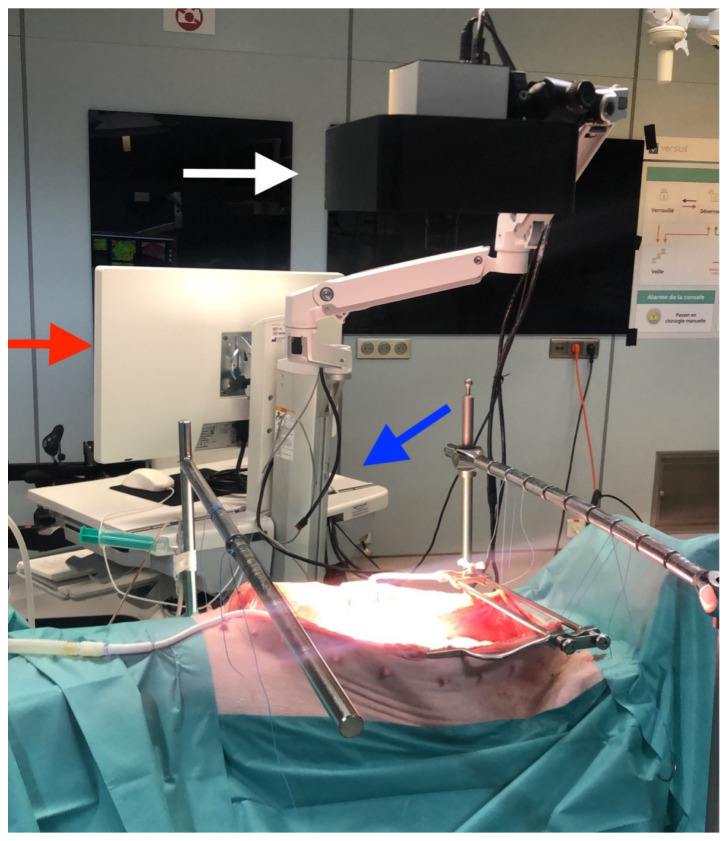
Set-up during experiments: The abdominal cavity was accessed via a midline laparotomy to expose the regions of interest (i.e., sigmoid colon, sigmoid mesocolon, and retroperitoneum). The distal lens of the hyperspectral camera was positioned 50 cm above the ROIs. External light interference was avoided during image acquisition. The TIVITA™ tissue camera is composed of a lightning unit (white arrow), a medical cart (red arrow), and a Box PC (blue arrow).

**Figure 2 diagnostics-12-02225-f002:**
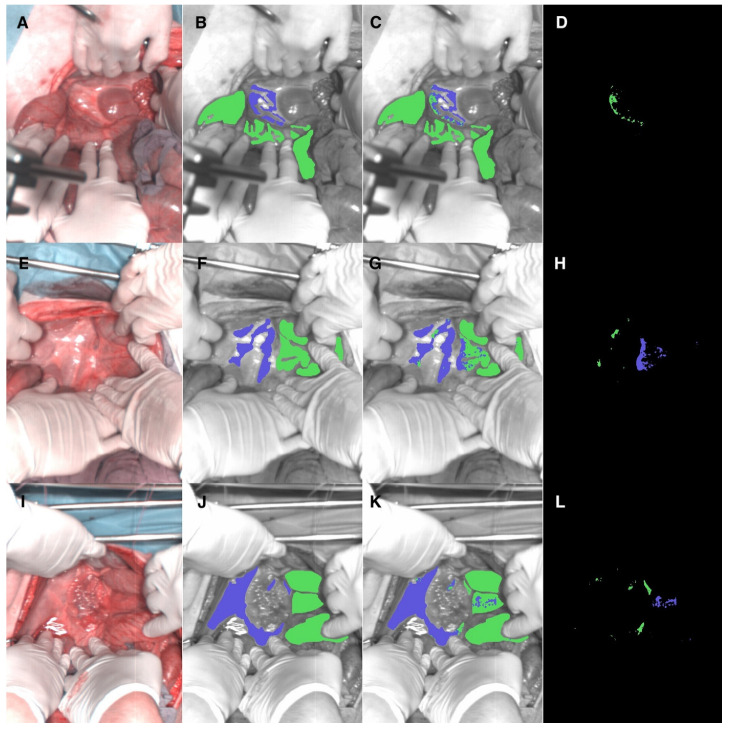
Example visualization of HS images, annotations, and automatic tissue recognition results provided by the trained CNN model. The figure is arranged in three rows of images, each corresponding to three example images from the dataset. From left to right, the first image column (**A**,**E**,**I**) shows the RGB images extracted from HSI data. The second column (**B**,**F**,**J**) shows the annotations provided by the expert surgeons. Purple corresponds to retroperitoneum (tissue to be preserved), and green corresponds to colon-mesocolon (tissue to be resected). The third column (**C**,**G**,**K**) shows the automatically predicted tissue classes within the annotated regions provided by the CNN. The fourth column (**D**,**H**,**L**) shows the corresponding error maps (showing all misclassified pixels).

**Table 1 diagnostics-12-02225-t001:** Details of CNN architecture.

Layer	Filter Shape	Number of Output Channels	Stride	Number of Trainable Parameters
Conv1	(3.3.3)	20	(1.1.1)	560
ReLU	/	/	/	/
Pool1	(3.1.1)	20	(2.1.1)	1220
Conv2	(3.3.3)	35	(1.1.1)	18,935
ReLU	/	/	/	/
Pool2	(3.1.1)	35	(2.1.1)	3710
Conv3	(3.1.1)	35	(1.1.1)	3710
ReLU	/	/	/	/
Pool3	(2.1.1)	35	(2.1.1)	2485
ReLU	/	/	/	/
FC	(455.2.1)	2	/	912

Total trainable parameters: 31,532; The CNN had three convolutional layers (Conv1, Conv2, and Conv3) and three pooling layers (Pool1, Pool2, and Pool3). All three pooling layers were implemented as 1D convolutional filters (filtering in the spectral dimension), and they included a stride of 2 in the spectral dimension and a stride of 1 in the two spatial dimensions. The stride had the effect of reducing the spectral dimension by a factor of 2 at each pooling layer.

**Table 2 diagnostics-12-02225-t002:** Model performance metrics.

Mean ± SD	Recall (Sensitivity)	Specificity	F1 score	MCC	ROC AUC
**Tissue to be resected**	0.86 ± 0.16	0.79 ± 0.21	0.90 ± 0.11	0.60 ± 0.23	0.92 ± 0.12
**(Colon-Mesocolon)**					
*n* = 20					
**Tissue to be left**	0.79 ± 0.21	0.86 ± 0.16	0.65 ± 0.25	0.60 ± 0.23	0.92 ± 0.12
**(Retroperitneum)**					
*n* = 20					

Sensitivity, specificity, F1 score, MCC, and ROC AUC. Averaged metrics computed using macro-averaging (at image level) with standard deviation; MCC: Matthews correlation coefficient; ROC AUC: Receiver Operator Curve Area-Under-Curve.

## Data Availability

The datasets generated or analyzed during the current study are included in this published article.

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
