# Peer review of "Computer-Assisted Differentiation between Colon-Mesocolon and Retroperitoneum Using Hyperspectral Imaging (HSI) Technology"

_diagnostics, 2022, doi:10.3390/diagnostics12092225_

Round 1

Reviewer 1 Report (Previous Reviewer 1)

I think the revised version of this manuscript (ID: diagnostics-1919652) is OK for publication in its present form.

Author Response

Thank you for your critical appraisal, encouraging words and useful comments.

Reviewer 2 Report (Previous Reviewer 2)

Thank you for improving the manuscript.
I only have small points:
1. Please add a statement if or if not you have performed white balance alignment before capturing.
2. Please do some rounds of proofreading! E.g., first sentence in Discussion, a 'of' is missing. You have mixed up with , and . etc...
3. Reference 59 and 60 are equal; Reference 3 includes excessive '3.'

Author Response

  1. Thank you for your comment, we have added a supplementary explanation in The section 2.2. Anatomical relevance, surgical procedure, and hyperspectral data acquisition as follows; In this study, relative values were considered no white balance was performed during image acquisitions.

  1. Thank you very much for your comments. We have revised the manuscript with our institution’s several medical translators and proofreaders whose first language is English.

  1. We have made the corrections you indicated. Thank you very much for your point out.

This manuscript is a resubmission of an earlier submission. The following is a list of the peer review reports and author responses from that submission.

Round 1

Reviewer 1 Report

In this work, deep learning and a convolutional neural network (CNN) were combined to process hyper-spectral imaging (HSI) data, helping distinguish the colon and its mesentery from the retroperitoneum. Furthermore, it is useful for the complete mesocolic excision (CME) operation. I think this study is interesting and helpful for the colorectal surgery. Thus, acceptance for publication is recommended after completing the following minor revisions:

  1. By using the authors’ technology, is the success rate of CME surgery increased? The authors should give the related results and discussion.
  2. Please have a space between number and unit.
  3. In Figure 1, the accurate information should be given with arrows.
  4. Please supplement the explanation of Figure 2C&D in the main text.

Reviewer 2 Report

The manuscript describes an experimental study with pig models evaluating binary tissue classification with HIS and CNN.

The overall presentation of the manuscript is ok. However, the manuscripts holds some typos and grammatical errors. Further, the tables are wrongly references: in Sec. 2.4.2, there is written “Table 1 in the supplements”, but there is no supplements and I think table 2 is meant; in Sec. 3.2. is a reference of Tab 2, but I guess it need to be Tab 1. Some references hold the reference number in the description as well.
Hence, some rounds of proofreading are needed!

In terms of content, I have the following major comments and questions:

  1. I recommend to specify the camera settings in more detail. In sec 2.2, the spectral resolution is not specified. A spectral range 500-1000 in 5nm steps would result in 101 bands, however, in Sec. 2.4.3 the HSI sub-cube is specified as 5x5x100. Thus, I suppose that the camera has 100 bands.
  2. Does the camera needs a calibration process? If so, please specify!
  3. The content of the annotation data is never been described. There are two classes, but how many pixels were annotated for each class? It there a bias or are both classes equally distributed?
    In Fig. 2, for example, it looks like there an imbalance to colon tissue (green annotation). How was this imbalance handled?
  4. Network: Please be specific in the text for the description of the network, in Sec. 2.4.2, 3 conv layer and 3 pooling layer. What kind of pooling are you using?
  5. Training/Evaluation: Had the evaluation performed only on annotation pixels or on all pixels of an HSI image? What happened with undecided/unclear decisions?
  6. Have you borders of annotations in your 5x5 patches? If so, how have you handled that?
  7. Results: Sec. 3.1, the last statement in that section is nonsense. As the image holds mostly non-annotated pixel, clearly the error map is mostly black. That is self-fulfilling. You could present the portion of wrongly specified compared to all blue labels, or so.
  8. F1 score of retroperitneum, i.e. tissue to be left is not so good, which correlates to Fig. 2D. But this has not been discussed somewhere in the manuscript. That should be caught up.
  9. Confusion matrix is not needed, as it holds no additional information. Sec 3.3 can be deleted. You had a binary task with two classes, the confusion matrix is just your sensitivity/specificity metrics, which already have been presented in sec. 3.2
  10. Discussion: The discussion has to be re-written! The first 4 paragraphs, i.e. the entire text from the beginning until “The potential clinical relevance of HSI lies in the better …”, are content for an introduction and mainly uninteresting for that manuscript. Only 2-3 sentences are of interest and need to be shifted to the introduction or material and methods sections:
    “Based on these results, we decided to adopt the CNN model …” -> move to MM
    “The AI automatic recognition of colorectal tumors is still in its infancy, …” -> move to Intoduction
    “In the present in vivo non-survival porcine model, …” -> can be remain in Discussion
    “Interestingly, the mesentery and … overall tissue spectral profile [60].” -> some parts of this paragraph can moved to MM, surgical procedure or annotation

    The rest of the discussion is ok, but only covers some parts of the study. The profundity of the discussion must definitely be expanded.